# Improving household surveys and use of data to address health inequities in three Asian cities: protocol for the Surveys for Urban Equity (SUE) mixed methods and feasibility study

Helen Elsey,[1] Ak Narayan Poudel,[1] Tim Ensor,[1] Tolib Mirzoev,[1] James Nicholas Newell,[1] Joseph Paul Hicks,[1] Christopher Cartwright,[1] David Wong,[1] Caroline Tait,[1] Sushil Baral,[2] Radheshyam Bhattarai,[2] Sudeepa Khanal,[2] Rajeev Dhungel,[2] Subash Gajurel,[2] Shraddha Manandhar,[3] Saidur Mashreky,[4] Junnatul Ferdoush,[4] Rumana Huque,[5] Tarana Ferdous,[5] Shammi Nasreen,[5] Hoang Van Minh,[6] Duong Minh Duc,[6] Bao Ngoc,[6] Dana Thomson,[7,8,9] Hilary Wallace[10]

For numbered affiliations see end of article.

**Correspondence to**
Dr Helen Elsey;
h.elsey@leeds.ac.uk

## ABSTRACT

**Introduction** As rapid urbanisation transforms the sociodemographic structures within cities, standard survey methods, which have remained unchanged for many years, under-represent the urban poorest. This leads to an overly positive picture of urban health, distorting appropriate allocation of resources between rural and urban and within urban areas. Here, we present a protocol for our study which (i) tests novel methods to improve representation of urban populations in household surveys and measure mental health and injuries, (ii) explores urban poverty and compares measures of poverty and 'slumness' and (iii) works with city authorities to understand, and potentially improve, utilisation of data on urban health for planning more equitable services.

**Methods and analysis** We will conduct household surveys in Kathmandu, Hanoi and Dhaka to test novel methods: (i) gridded population sampling; (ii) enumeration using open-access online maps and (iii) one-stage versus two-stage cluster sampling. We will test reliability of an observational tool to categorise neighbourhoods as slum areas. Within the survey, we will assess the appropriateness of a short set of questions to measure depression and injuries. Questionnaire data will also be used to compare asset-based, consumption-based and income-based measures of poverty. Participatory methods will identify perceptions of wealth in two communities in each city. The analysis will combine quantitative and qualitative findings to recommend appropriate measures of poverty in urban areas. We will conduct qualitative interviews and establish communities of practice with government staff in each city on use of data for planning. Framework approach will be used to analyse qualitative data allowing comparison across city settings.

**Ethics and dissemination** Ethical approvals have been granted by ethics committees from the UK, Nepal, Bangladesh and Vietnam. Findings will be disseminated through conference papers, peer-reviewed open access

## Strengths and limitations of this study

► Multisite study in three Asian cities testing novel survey methods to improve representation of urban poor (leading to reduced selection bias).
► Mixed methods design allowing comparison of wealth measures appropriate to the urban poor (leading to improved wealth classification).
► Testing of a simple area-based measure of 'slumness' that can be used in future surveys and censuses (leading to improved slum area classification).
► The scale of the study in two of the cities (Dhaka and Hanoi) is insufficient to estimate prevalence of depression and injuries.

articles and workshops with policy-makers and survey experts in Kathmandu, Hanoi and Dhaka.

## INTRODUCTION

In low-and middle-income countries (LMICs), household surveys—such as Demographic and Health Surveys (DHS)[1] and WHO's Step-wise Approach to Surveillance (STEPS)[2]—provide vital data sources to inform the health sector nationally. The methods used in these surveys have become standardised, allowing valuable comparisons in over 100 countries and over time, for >30 years.[3] However, increased population mobility and rapid and unplanned urbanisation means that an increasing proportion of urban dwellers live in unplanned, unregistered settlements, in non-standard living quarters (such as a hostel, shop or guesthouse), and in non-family living

BMJ

arrangements (such as a group of flat-mates or multi-family dwellings). There is an urgent need for survey methods to keep pace with these changes. Current methods used in surveys such as DHS, Multiple Indicator Cluster Survey, Global Adult Tobacco Survey, the WHO STEPS do not allow analysis of interurban and intraurban health differences and systematically under-represent the urban poor.[1–4] This means that health issues experienced by the urban poor are masked by better health outcomes among the urban wealthy,[4 5] and severely limits the ability of policy-makers in LMICs to take decisions on resource allocation between rural and urban areas, between population groups and localities within urban areas. This limits actions to target health programmes, both preventative and curative, at the most disadvantaged and fuels inequities within urban areas and nationally.

There are several methodological reasons why the urban poorest are under-represented in surveys. First, census data, which is used to select first-stage samples, is often outdated and undercounts informally settled households.[6] Second, by design, surveys typically exclude the homeless and institutional populations. Use of two-stage cluster sampling methods requires two visits to households over several months or years, resulting in underlisting or higher non-response by mobile and fragile households. Third, underlisting and undersampling of poorer households can occur if standardised, detailed protocols are not used by enumerators to interact with residents during the household listing process. For example, multihousehold dwellings will be underlisted if the enumerator assumes one dwelling to be occupied by one household or poorer members of households, such as guards and servants may be excluded. Furthermore, peri-urban communities[7] frequently home to urban migrants and slum areas, maybe classified as rural. These factors all lead to underestimation of the informal urban population and therefore underestimation of the numbers in poverty in urban areas.[8]

Even though surveys such as DHS and STEPS have large sample sizes (between 5000 and 30 000 households), they are not designed to allow valid interurban or intraurban comparisons. Analysis illustrates that there are clearly too few of the poorest households in urban areas included to make estimates and comparisons. For example, the 2011 Nepal[9] and 2011 Bangladesh DHS[10] only recorded 168 and 515 individuals, respectively, from the bottom wealth quintile, far below the 1500 sample required to make estimations of indicators of interest.[1]

Several methodological innovations offer promising solutions to improve the representation of the urban poor in household surveys. The first of these is the use of WorldPop[11] data with gridded population sampling to overcome limitations of outdated and incomplete census data. WorldPop population data provide open-access estimates of the number of people living in 100 m×100 m grid cells for all LMICs. The estimates are based on the most recent and detailed official population data available, usually published as a count per geographic unit. The population in each geographic unit is disaggregated into 100 m×100 m areas using a random forest model based on spatial covariates such as night-time light intensity, the distance to roads and other infrastructure and land cover type.[12] Gridded population estimates are available at a much finer scale than census enumeration area data, and can be updated using the most recent spatial covariates.

The second innovation is the complete enumeration of both dwellings and households in all sampling areas using an OpenStreetMap (OSM) Android application[13] and use of a detailed, standardised household mapping-listing protocol. OSM is an open-access online map of buildings, roads, rivers and landmarks, based on open-access data (including manually inputted, or crowdsourced, data). All major cities are mapped to some extent, and an increasing number of secondary cities are mapped in OSM. The third innovation is the use of one-stage sampling, which eliminates the time delay between listing and interviewing households.

The content of the questionnaires and measures used must also remain relevant to rapidly urbanising LMIC contexts. Of particular concern are current methods for categorisation of urban populations by wealth quintile. Wealth quintiles derived from DHS, STEPS or other large surveys are generally based on household physical assets such as type of water source, roof, floors and wall[14] and possessions such as a motorbike. The current system categorises urban dwellers living in formal buildings with solid floors, walls and roofs into a higher wealth quintile. However, better housing may mean higher rents so that households have less discretionary income to spend. Conversely, slum dwellers with insubstantial housing (which would put them in the lowest quintile) may pay little rent and have relatively greater financial resources.[15] Furthermore, wealth includes income, saving, access to credit and other financial assets beyond physical assets, but these are not currently accounted for in quintile development. Until measures can take into consideration the nature of vulnerability of the urban poor, wealth categorisations are unlikely to adequately classify households.

Increasingly, evidence shows that the urban poor are particularly vulnerable to key non-communicable diseases (NCDs) that are rarely collected within household surveys. Mental health and injuries are two of these neglected non-communicable diseases (NCDs); what little data there is demonstrates the negative impact of urban poor environments on mental health, fuelled by high levels of alcoholism, crime, gender-based violence and the fear of evictions and environmental hazards[16 17] and likewise, urban slums present high risks for injury, particularly among children,[18–21] for example, 43% of those under 18 years in Dhaka's slums had an injury in the last year[21] and road traffic accidents (RTAs) increased in South Asia by 22% between 1990 and 2013.[22] Furthermore, the effects of NCDs are disproportionately detrimental for the urban poorest as they struggle to meet the costs of care for these chronic diseases.[23 24] While DHS and STEPS NCD Risk Factor surveys provide valuable

national data on many NCDs (particularly cardiovascular disease and diabetes) and risk factors (diet, exercise, alcohol and tobacco use), mental ill-health and injuries, which cause high morbidity among the urban poor, are frequently not covered.[1 2] There are notable exceptions, such as Bangladesh which conducts the Bangladesh Health and Injury Survey[25]; however, such an extensive stand-alone survey is resource-intensive to conduct and not a viable option for many countries. Data on mental health frequently focus on severe mental health issues and suicide,[26] whereas depression, a much more common condition, is not measured in national surveys in LMICs leading to a dearth of population estimates of the prevalence of depression.

Attention is now being paid to the effects of living in a slum area[27 28] over and above the effects of living in a slum household as defined by UN-HABITAT.[29] These area-level effects may interact with household characteristics to impact on health and well-being. However, there is currently no agreed definition of a slum area and no validated measure to assess 'slumness' of a neighbourhood area. This means that empirical work to assess the negative or positive effects of living in or near a slum neighbourhood cannot be conducted.

Improving the representation of the urban poor within household surveys is the first crucial step. However, unless policy-makers and local government officers can make use of these data to inform their decision-making and monitoring of urban health, then such data, even if high quality, will not impact on urban health or addressing inequities in cities. Use of evidence and data to inform planning decisions is weak across LMICs,[30–32] but this is particularly the case for urban municipalities, which have been consistently under-funded and overlooked by donors and governments.[33] There is an urgent need to find ways of presenting data clearly and accessibly to enable use by decision-makers to target resources and services to those who most need them. Currently, important findings from household surveys are hidden in wordy reports and health management information is hard to access. Inevitably, such data are not used to inform planning and management decisions such as where to locate health centres or which risk factors and groups to target through health promotion. In this paper, we share the methods we will use in a 2-year mixed methods study, followed by a discussion of the strengths, limitations and implications for policy and practice.

## Objectives

Our aim is to test the feasibility, cost and appropriateness of novel survey and visualisation methods to appropriately represent all wealth groups in urban areas. We will also identify and test questions to assess two neglected NCDs—depression and injuries—and develop urban-appropriate definitions of a household and measures of wealth. We will work with municipal governments to understand and use available data for urban planning.

Specific objectives are to:

▶ Pilot the gridded sampling, mapping and one-stage sampling methods to assess feasibility in terms of time, cost, skill of team and statistical efficiency.
▶ To identify and test the comprehensibility, acceptability and feasibility of a short suite of questions to enable understanding of the epidemiology of depression and injuries in urban contexts in LMICs.
▶ Identify appropriate measures of urban wealth and poverty for use within household surveys.
▶ Assess the extent and nature of data use in planning and management processes and practices within urban local governments.
▶ Engage closely with municipalities to develop appropriate and feasible data visualisation tools to support planning and management.
▶ Establish and strengthen collaborations with health research institutions, international and national bodies conducting cross-sectional household surveys, national and local government departments and academics to share expertise and build capabilities on survey design, assessment of wealth, mental health and injuries, data visualisation and use of data to address urban inequities.

## METHODS AND ANALYSIS
### Study design

Our study uses quantitative (survey), qualitative (in-depth interviews, focus group discussion, non-participant observation, literature reviews) and participatory methods (social mapping, transect walk, photovoice and wealth ranking) to answer the objectives above. Data will be collected from October 2017 to January 2019. We will use a sequential mixed methods design to enable findings from one method to inform further data collection and analysis.

### Study setting

The study will take place in three cities: Hanoi, Vietnam; Dhaka, Bangladesh and Kathmandu, Nepal. These cities have been selected as they display different characteristics of urban living (table 1). Furthermore, we have existing strong partnerships with health research institutions in these countries with experience in cross-sectional surveys and strong government links.

### Piloting gridded sampling, mapping and one-stage sampling methods

We will pilot the novel survey methods in Kathmandu before exploring the feasibility of their use in Hanoi and Dhaka; this will enable us to learn lessons and share training resources across our city teams. We will use gridded population sampling techniques, making use of WorldPop data to more accurately select clusters. Where applicable, we will use a modelled boundary called the Global Human Settlement Layer—City Model (GHS-SMOD)[34] rather than official administrative boundaries to define the survey coverage area to ensure that both formally and informally settled populations, including those beyond

| Table 1 | Characteristics of study settings | | | |
|---|---|---|---|---|
| **Key indicators** | | **Nepal** | **Bangladesh** | **Vietnam** |
| Income level of country (per capita GDP)[76] | | US$722 (2016) | US$1355 (2016) | US$2171 (2016) |
| Study city population[76] | | 1 181 000 (2015) | 17 598 000 (2015) | 3 629 000 (2015) |
| Rate of growth of urban population[77] | | 3.18%** | 3.55%** | 2.95%** |
| Maternal mortality (per 100 000 live birth)[78] | | 258 | 176 | 54 |
| Under 5 mortality (per 1000 live birth)[78] | | 35.8 | 37.6 | 21.7 |
| Vaccine coverage among those aged 1 year (%)[78] | | 91 | 94 | 97 |
| Stunting among children (<5 years) (%)[78] | | 37.1 | 36.1 | 24.6 |
| Death from RTA (per 100 000)[78] | | 17 | 13.6 | 24.5 |
| Suicide mortality rate (per 100 000)[78] | | 6 | 5.5 | 7.4 |
| Mortality due to unsafe WASH (water, sanitation and hygiene) services (per 100 000)[78] | | 12.9 | 6 | 2 |
| Safely managed drinking water services (%)[78] | | 92 | 87 | 98 |
| Safely managed sanitation services (%)[78] | | 46 | 61 | 78 |

the centre-city administrative boundaries, are included (eg, see figure 1 of Kathmandu Valley, Nepal).

The enumeration team (responsible for mapping all structures including tents and listing all households) will be trained to use a script to list the structures or buildings, levels, dwellings and households for every structure in the sampling areas, including non-residential structures where guards, cleaning staff and other people normally stay. We will map the urban populations in these clusters using OSM as our base map, adding or updating buildings within selected areas using OSM's built-in iD Editor, thus making the data publicly available. The methods will be piloted as part of the Kathmandu household survey and the experience shared with teams in Hanoi and Dhaka.

We will aim for a sample size of 1200 in the Kathmandu survey, enabling us to estimate key depression and injury indicators with a maximum margin of error of ±4.27% with 95% confidence (assuming the most conservative scenario where an indicator is estimated at 50%). This assumes a design effect of 1.41 (the mean design effect across all indicators for men and women in urban areas in Nepal DHS 2011),[9] a household and an individual response rate of 0.98 and 0.93, respectively (based on conservative estimates from response rates in urban areas in Nepal 2011 DHS) and one eligible individual per household. This sample population will be distributed across 60 clusters in the Kathmandu. This approach will allow estimates of prevalence of depression and injury.

### Survey participants

We will interview adults, 18 years and above who provide informed consent and are not under the influence of any substances or mentally unable to respond. We will use the DHS definition of a household head.[35] Within one-stage sampling, we will conduct a separate questionnaire with non-relatives and staff staying within a household. Within two-stage sampling, these would be included within the main household questionnaire. In one-stage sampling, we

will also conduct the questionnaire with residents of over 7 days in any hostel or guesthouse that is not solely for social/healthcare (such as an older peoples' home) or education (college dorm).

To compare the effectiveness of one-stage sampling compared with two-stage sampling, in the Kathmandu survey we will randomly allocate half of the clusters to each approach. The two-stage clusters will have approximately 200 households each. One-stage sampling of approximately 20 households in each sampling area will be facilitated by the use of WorldPop 100 m×100 m grid cells rather than much larger census enumeration areas as the sampling frame.[36] The enumeration teams will map and list dwellings (not households) in each one-stage sampling area, and segmentation will be used if needed, to ensure that interviewers can feasibly list and interview all households at a later visit. Counts of homeless people and interviews with long-term residents of guesthouses will additionally be performed in one-stage sampling areas.

In the two-stage sample, households will be selected using random interval sampling based on the prior enumeration of households following the methods used in DHS and similar surveys.[37] In the one-stage sample, interviewers will be trained to identify all households in the cluster, including the homeless, residents in hostels and individuals such as servants and guards, and approach them for interviewing. This will allow inclusion of unconventional households such as those staying in shops or guesthouses for over a week who would not have been captured in the traditional enumeration process used in two-stage sampling.

In Dhaka and Hanoi, 400 households will be sampled and only one-stage sampling will be conducted. While this sample size will allow feasibility testing of the novel methods, it will not allow estimates of prevalence of depression or injury.

In Dhaka, two urban neighbourhoods will be purposively selected to illustrate both mixed (poor and

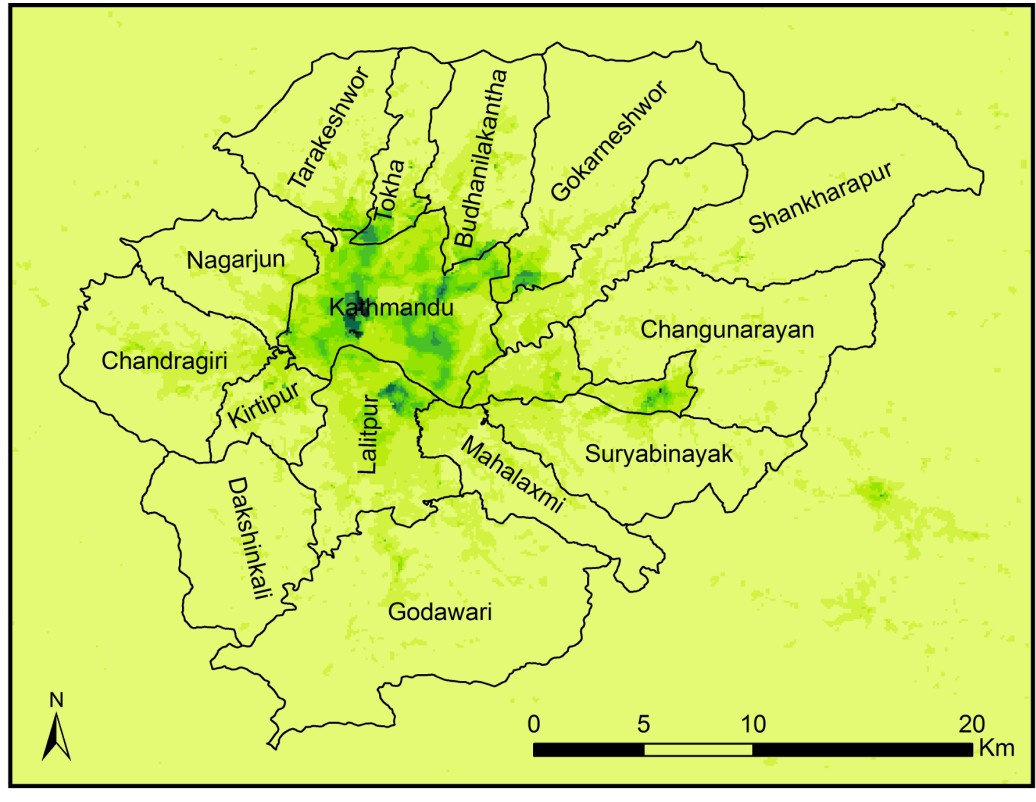

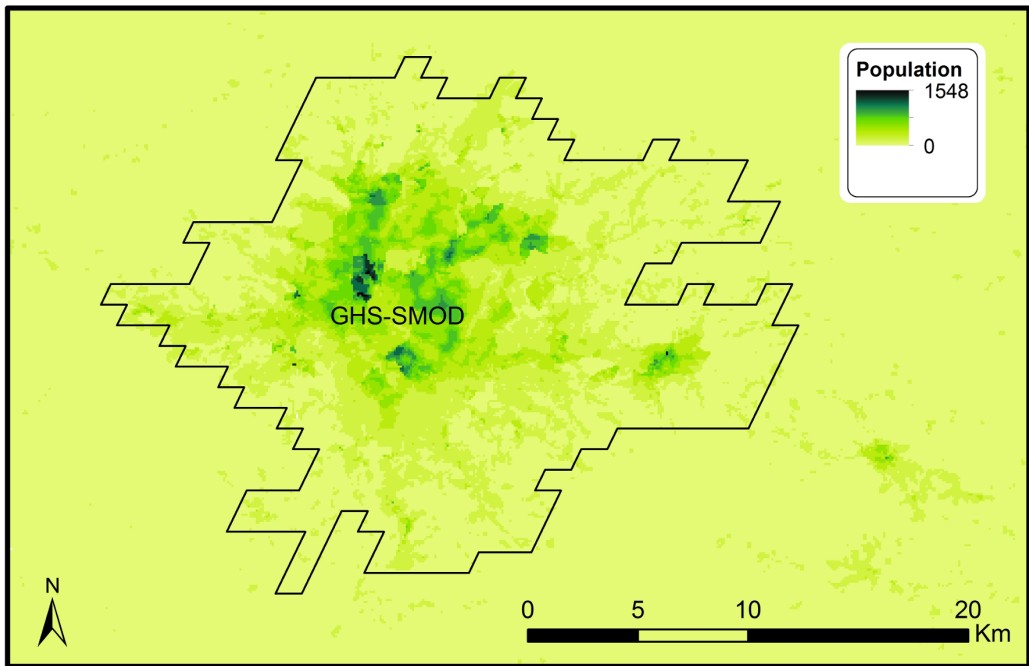

**Figure 1** WorldPop population estimates (2017), municipality boundaries and Global Human Settlement Layer—City Model (GHS-SMOD) 'dense urban' area boundary.

middle-income) households and informal slum settlements for the feasibility testing. The mixed neighbourhood which covers the administrative area of a ward, will have 15 main and 5 backup clusters. The informal settlement covers a smaller area and will have five main clusters and two backup clusters. Therefore, there will be 20 main clusters and 7 backup clusters in Dhaka. As the population density of Dhaka is very high, we will have cluster of 100 m×100 m each. Likewise, Hanoi will have 20 main clusters and 10 backup clusters. In Hanoi, each cluster will be of 200 m×200 m size as population density is lower than Dhaka. Backup clusters are selected because, with the use of the WorldPop dataset, it is likely that some of the selected clusters will have little to no population. Around 20 households will be randomly selected for interview from each cluster.

Detailed step-by-step guides for the use of the novel methods will be developed for the survey planning team and the enumerators and in-depth training will be provided. In Nepal, this will be provided by coauthor DT. The Dhaka and Hanoi teams will then receive training from the Nepal team with additional support from DT as needed. A detailed manual to guide interviewers in the delivery of the questionnaire will be developed and adapted for each city questionnaire. The methods manuals and questionnaires will be available on the study website (https://medhealth.leeds.ac.uk/info/691/research/2388/sue). The interviewers will receive 5 days training to ensure they are confident in the survey protocol and delivery of all sections of the questionnaire.

## Public and patient engagement

The research question was informed by our previous work with poor urban communities, particularly those in informal settlements, who have expressed a sense of powerlessness to influence their living conditions and inaccessibility of public services.[15] While no poor urban communities were involved in the development of the protocol for this study, the participatory methods we have chosen will maximise their engagement and enable their perspectives of the characteristics of poverty to inform our analysis of the most suitable measures of poverty for urban areas.

## Data collection

### Feasibility of gridded sampling using WorldPop data

In each city, we will test the effectiveness of the gridded sampling approach using WorldPop data to identify enough clusters of the target population size. This will be evaluated in terms of number of clusters that were dropped and replaced because no buildings were visible in satellite imagery before fieldwork, or no dwellings/households were identified during fieldwork (eg, buildings belonging to a factory or temple where no one stays overnight). Segmentation of very large clusters will be used as is common in standard surveys. However, gridded population sampling may be more likely than standard surveys to result in clusters with considerably smaller population than desired. We will track the number of low-population clusters.

### Feasibility of using OpenStreetMap for enumeration

We will record the time taken and resources, including equipment, transport and salaries of staff, needed to map and list the households using OSM. The skills and extent of training needed for enumerators to use the software to use OSM will also be recorded. Where data are available, we will compare these findings with DHS reports from each of the cities. We will also record any challenges using OSM, for example, where insufficient information has been added to OSM through crowd-sourcing, negating its use as a viable approach to mapping. Any other programmes used will be documented and assessed.

### Feasibility of one-stage versus two-stage sampling

In Kathmandu where both one-stage and two-stage sampling will be conducted, we will compare the time taken, resources and skills needed between the two sampling approaches. In each city, following the mapping and listing process, a focus group discussion will be held with the enumerators to understand any practical constraints and issues arising in the enumeration using OSM. They will also be asked to reflect on any differences they found in implementing the enumeration process in one-stage and two-stage sample areas. Focus groups will be audio-recorded and analysed thematically.

### Questionnaire development

The questionnaire working group will be established with members from each city team. Existing, where possible, validated questionnaires will be selected to cover the domains of the questionnaire as outlined in table 2. The generic questionnaire will then be adapted to ensure appropriate terminology for each city context and translated into the main national language: Nepali, Bengali and Vietnamese. All questionnaires will be available in English at: https://medhealth.leeds.ac.uk/info/691/research/2388/sue.

We will conduct a focus group discussion with interviewers in each city to understand the feasibility and acceptability of the depression and injury sections of the questionnaire; during these focus groups, the interviewers will reflect on respondents' ease in understanding and answering the questions. Interviewers will be asked to discuss experiences of interviewing 'non-traditional' households such as street-sleepers or long-term guesthouse residents as part of the one-stage sampling approach. They will also share feedback on the use of the 'Sample Area Observation Form' as a way of classifying the slum-like nature of the area. All focus groups will be audio-recorded and analysed thematically.

### Understanding wealth and poverty from the perspective of urban poor communities

To explore the characteristics of poverty and vulnerability from the perspective of community members, and to understand the different types of household, we will use a selection of participatory methods in two poor neighbourhoods in each of the three cities. The neighbourhoods will be selected to illustrate different types of urban poverty, for example, communities with poor living next to better-off households and informal settlements. The methods to be used are social mapping, transect walk, photovoice and wealth ranking. The exact order and conduct of the methods will vary as appropriate to the context, but is likely to involve an initial social mapping exercise with community leaders showing key features of the area and identifying any clusters of poor households, followed by a transect walk around their community identifying different types of household and explaining the causes of poverty and

**Table 2** Questionnaire sections, respondents and purpose of each section

| Questionnaire section | Respondent (aged ≥18 years) | Purpose | Topics covered and source of questions |
|---|---|---|---|
| Household questionnaire | Household head/most knowledgeable about household | ► To estimate sociodemographic characteristics, number of injuries per household, household migration patterns and social capital. <br> ► To compare measures of wealth/poverty/slum/non-slum household. <br> ► To identify those injured in last 6 months for individual injury questionnaire. | ► Sociodemographic characteristics of household members: age, gender, education level, caste/religion/ethnicity, occupation.[1] <br> ► Household members with a disability[79] and all those injured in the last 6 months.[25] <br> ► Household assets.[1] <br> ► Slum/non-slum household definition.[80] <br> ► Progress out of poverty index questions.[42] <br> ► Consumption.[43] <br> ► Income (tailor made—building on qualitative findings). <br> ► Migration.[81] <br> ► Social capital.[80] |
| Individual questionnaire | Randomly selected from all household members using the Kish method[82 83] | ► To estimate the prevalence of depression (Kathmandu only). <br> ► To assess the acceptability of PHQ9 and somatic questions. <br> ► To assess level of agreement between PHQ9 scores and somatic symptoms. <br> ► To explore associations between mobile phone ownership, migration and social capital. | ► PHQ9 (Patient Health Questionnaire 9).[48–50 84–86] <br> ► Somatic symptoms of mental ill-health (developed by national mental health experts in each country). <br> ► Affect questions from Washington Group Extended Set (Hanoi and Dhaka only).[79] <br> ► Migration.[81] <br> ► Social capital.[80] |
| Individual questionnaire | All those injured in the last 6 months and those died in last 1 year | ► To estimate the prevalence of injuries (Kathmandu only). <br> ► To assess the acceptability of a short set of injury questions. | ► Cause, nature and impact of injury, including injury-related death.[25 62] |
| Sample area observation form | Two members of the research team, independently, after completion of household survey in a cluster | ► To compare a simple subjective categorisation of 'slumness' with a list of key slum characteristics. <br> ► To assess the level of agreement between the two researchers. | ► Simple 'slumness' categorisation of the sampling area: (i) non-slum, (ii) slum, (iii) mixed, (iv) distinct slum and non-slum. <br> ► Characteristics of slums: 17 questions on social and environmental risks, eg, built on undesirable land due to slope, flood zone, crime. <br> ► Lack of facilities/infrastructure eg, absence of services, eg, health, education, clean fuels and technologies, transportation. <br> ► Unplanned and disorganised settlement eg, nature of roads and buildings. <br> ► Contamination, eg, extent of garbage/waste, open defecation and air, land and water pollution.[80] |

vulnerable in their community. Community members will take photographs of dwellings in the poorest and wealthiest categories. These will then be used in group discussions with participants categorising households into poor, not-so-poor and better-off households. Facilitators trained in participatory methods will lead the discussions probing to understand community perspectives on wealth, vulnerability and poverty. Where appropriate, these methods will be held with separate groups of men and women. The discussions accompanying all the methods will be audio-recorded and transcribed. 'Framework approach'[38] will be used to analyse the data to enable understanding of the wealth categories. The findings from these participatory methods will inform the choice of questions in the questionnaire to ensure that, where possible, multiple dimensions of urban poverty are measured within the survey.

## Assess the extent of data use in planning and management processes and practices

In each of the three cities, we will build an understanding of the extent of use of data in health planning processes and practices within local government bodies. It should be noted that in the three study countries, local governments play a major role in health, including in some cases, the management of health facilities in the city. We will use three qualitative methods: in-depth interviews (IDIs), non-participant observations and document review. The interviews will explore stakeholder practices and experiences with health planning, identify which data are used in health planning and management and how and explore stakeholder preferences for the presentation of data to enhance its use in urban health planning and management processes. We estimate that approximately 10 IDIs in each city will allow us to develop a good understanding of the current practices and explore stakeholder experiences.

We will also conduct non-participant observations of key health planning events (eg, joint annual reviews or health planning and budgeting meetings) in each of the three cities to allow us to understand the processes involved in group decisions, for example, in relation to prioritisation and allocation of resources, and to triangulate findings from the IDIs. Observations will focus on what, if any, data are used to inform decisions, how they are presented and interpreted and how they influence decisions. Reviews of health planning documents will help us understand how the use of data is formally documented and will allow comparisons with findings emerging from IDIs and observations.

Participants for these interviews will be selected purposefully, based on roles in urban health and health-related planning processes. Stakeholders involved in IDIs in each city will include urban decision-makers from local government, and key individuals from central government and from international non-governmental organisations who have a stake in the urban planning.

## Engage closely with municipalities to develop data visualisation tools

Building on the findings and relationships established through the qualitative work with key urban planning stakeholders, we will work with municipality and central government decision-makers in Kathmandu, Dhaka and Hanoi to establish communities of practice (CoP).[39] These CoPs will include decision-makers, data analysts and planners. We envisage between three and six meetings over a 6-month period. The CoPs will explore the most useful ways of presenting data to enable their use in local planning, management and monitoring. In particular, we will explore the possibility of using existing open-source platforms such as DHIS2.[40] This process of continual and careful engagement with the 'end-users' of the data will ensure their needs inform any adaptations to visualisation programmes. The issues raised and proposed solutions identified during the CoP meetings will be recorded and analysed thematically to identify lessons learnt on the most useful means for supporting data use within local city governments.

## Analyse our sampling methodology and questionnaire data

Within the Kathmandu sample, we will compare one-stage versus two-stage sampling in terms of their logistical (cost and time required, as well as number of enumerators and interviewers and their required skills) and statistical efficiency to allow us to evaluate the relative benefits and costs of each method. We will assess statistical efficiency by calculating intracluster correlation coefficients (ICCs) for a small range of key outcomes taken from key domains of the questionnaire, separately for one-stage and two-stage samples, to explore whether the two sampling methods differ in their statistical efficiency depending on outcome type. We will also calculate ICCs for all main outcomes that have been collected from comparably high proportions of respondents for one-stage and two-stage samples, and compare them using a t-test to compare the overall statistical efficiency of each method. In the other city samples, we will report the cost, time and skills of enumerators and interviewers needed in the one-stage sample design. The different types of household identified by each sampling approach will also be analysed and in the case of Kathmandu, compared with two-stage sampling. We will compare the inter-rater reliability of two researchers in designating PSUs as slum/non-slum/mixed via their completion of the 'Sample Area Observation Form', using Cohen's κ in a probabilistic benchmarking procedure.[41]

From our survey, data we will present key outcomes as percentages (for categorical variables) and means (for continuous variables) with their estimated 95% CIs. We will explore how depression and injury occurrence vary in relation to a range of likely influences at the individual and household level, as well as at the area-level via the slum neighbourhood effect, and between countries, using multiple linear and logistic regression models. We will also compare measures of monetary, non-monetary economic status and vulnerability to understand how their relationships vary across different types of urban settlements and households. All composite indicators will be created according to pre-existing stipulated and published analysis methods.[42–46] We will compare the components of these indicators with the factors identified by urban communities during the participatory methods to identify any missing components of common measures of wealth and vulnerability. All survey data analyses (as necessary) will account for the complex features of the surveys' designs by adjusting for (in the case of Nepal) stratification between one-stage and two-stage sampling schemes (and for all countries), clustering due to multi-stage sampling and unequal selection probabilities (via sampling weights).[47]

We will also assess the validity of using the somatic questions as a means of identifying depression in this population, based on taking the PHQ9 as the gold standard of

diagnosis, using a cut-off score of 10 based on the current evidence from these countries and similar contexts[48–50] (or modified in light of any new evidence). We will assess validity in terms of a number of factors including: (1) acceptability based on completeness of responses, (2) distributional properties of the scores based on skewness and floor and ceiling effects), (3) internal consistency based on McDonald's and Chronbach's α, (4) dimensionality based on an exploratory factor analysis, (5) convergent validity between the somatic question scores and key measures of 'life difficulties' including disability, non-working patterns and low social capital (using suitable correlation measures), (6) criterion validity based on a receiver operating characteristic curve analysis to determine optimal cut-off score(s), reporting the area under the curve, sensitivity, specificity, positive and negative predictive values, percentage correctly classified and Youden's index.

### Qualitative data analysis

Results of data collected from multiple qualitative methods need to be compared across the three different countries. Therefore, a framework approach[38 51] will be used to analyse the qualitative data obtained from different qualitative and participatory methods. This approach will allow us to apply predetermined themes, or a framework, to structure the analysis and ensure comparability of results across the different cities, while allowing for sufficient flexibility for new themes to emerge from the data.

### DISCUSSION

The inadequacy of existing survey methods in representing the urban poor and understanding their health needs is now well documented.[6 52–55] This undercounting of the urban poor is not merely a technical issue of survey methodology. As surveys provide the main data source for governments and donors to prioritise funding and develop strategy within all sectors of government, the political and resource implications cannot be overstated. Attempts have been made to quantify the extent of this under-representation. Carr-Hill estimates that 250 million transient, institutionalised and homeless people—including the urban poor—are missed worldwide from the sampling frames of surveys and censuses.[6] In Nairobi's Kibera slum, possibly the largest slum in Africa, well-designed studies report population estimates 18%–59% greater than those of Kenya's most recent national census.[55] Similar disparities between census and official estimates and more detailed mapping and enumeration studies of the urban poor have been documented in both India[53] and Egypt.[52] Cost-efficient methods to address this underestimation of the urban poor in sampling frames are urgently needed.

A further issue limiting the inclusion of some of the poorest urban dwellers in surveys is the definition of a household itself. Standard definitions of a household emphasise permanent structures and closed households;

this creates particular issues in urban areas where those in temporary shelters, street-sleepers or those sleeping in places of work are likely to be omitted.[8] The constraints of measuring only those in 'bounded, largely impermeable units' has been recognised by Randall and Coast,[56] whose work on poverty in Tanzania and Burkina Faso instead identified 'closed' and 'open' households, where open households in urban areas encompass migrants and other incomers from rural areas. Finding ways of counting those within open households as well as the most vulnerable urban dwellers beyond any household structure is a key challenge in survey design. Some researchers working on urban surveys have recommended a move away from households as the unit of analysis, and adopt 'people and communities' as units.[57] The standard two-stage cluster sampling approach requires an initial listing of all households in the cluster and then the random selection of households for interview. Given the impermanence of open households, and other, types of urban 'households' that are commonly excluded from two-stage sampling, we hypothesise that one-stage sampling of all individuals in a smaller cluster will provide a means of collecting representative data from all urban dwellers, including vulnerable people.

There has been much debate on methods to measure and understand urban poverty and inequities.[52] The appropriateness of using consumption measures, often seen as the gold standard, within urban areas has been questioned due to their limited focus on meals eaten out and the increased likelihood of household members eating independently away from home.[58] The importance of expenditures on rent and household amenities within urban areas is also frequently underestimated within consumption questionnaires. For example, a study in Zambia found non-food needs in Lusaka that were 10 times higher than official estimates and this was primarily down to expenditure on housing.[59] As identified in a recent Overseas Development Institute report,[55] monetary poverty measures can misrepresent urban poverty, although the direction of the bias may not always be clear. Composite multidimensional measures are attractive, but it can be challenging to gather the data required to measure all dimensions, for example, the 'security of tenure' component of the UNHABITAT slum household definition or the nature of access to services and infrastructure.[55] Comparing individual data on several measures of poverty and contrasting these findings with qualitative information from poor urban residents in the three cities in this study should provide valuable insights to these issues.[54]

To change household survey methods, such as DHS, that have been used consistently over the last 30 years or more requires robust assessments of proposed novel methods, carefully identifying their feasibility, validity and costs. We aim to contribute to this growing evidence base of possible new methods.

Urbanisation brings with it a changing burden of disease. Cities require thoughtful planning in the local

use of resources to manage health determinants.[60] It is vital to ensure that routine household surveys are able to collect health and risk factor information of relevance to poor urban populations if responses to urban ill-health are to be appropriate. Injuries and poor mental health have been found to be associated with slum living.[17 18 61] However, the current lack of modules assessing injuries and common mental health complaints such as depression within large household surveys means that there is little epidemiological data available to inform urban health planners. Considerable work has been done to identify appropriate measures of injury, their cause and impact.[25 62] However, challenges remain in measuring depression given different cultural understandings[63] and taboos around mental health.[64] We propose that a symptom-based assessment of depression may provide a valid and feasible way of collecting population level data on depression in urban areas.

The epidemiological data that are available on mental health and injuries highlight the influence of area-level as well as individual-level and household-level risk factors. In Dhaka's slums, features of the natural environment, population density, flood risk and public sanitation as well as household (housing) and individual factors (job satisfaction and income generation) were found to be associated with poor mental well-being.[65] Such findings support recommendations for an area measure of deprivation or 'slumness'.[28 66] Developing such a measure is challenging given the lack of a global definition of a slum. We have drawn on the existing literature and discussion among an expert group drawn from UN agencies, national bureau of statistics, donors and academics to develop a global definition of slum areas.[67]

The final strand of the project, understanding the constraints that local governments are facing in identifying and using appropriate data to inform urban decision making,[68] is one of the most crucial elements of the study. However, robust and high-quality data on urban health, without the means for staff within city corporations and municipalities to use these data there is little chance of improvement in urban health and reductions in inequalities.[69–73]

## Strengths and limitations

This research has a number of strengths. First, it tests the feasibility and costs of novel survey methods (gridded sampling with WorldPop, OSM enumeration) in three Asian cities and explores benefits and drawbacks of one-stage and two-stage sampling in identifying some of the most vulnerable urban dwellers. Second, the research allows robust assessment of the appropriateness of quantitative measures of wealth commonly used in household surveys and comparisons with qualitative findings from poor urban communities. Third, it will enhance understanding of injuries and depression among urban populations, by recommending a simple, short set of questions that can be used within exiting household surveys to estimate prevalence. Fourth, it

will provide insights on practical ways to support local government decision-makers to use available evidence to inform their planning and monitoring to improve urban health.

While our study may not identify and specify all changes required to household survey, our pilot work will provide sound empirical data which can inform global efforts to improve the representation of the urban poor and quality of data collected to understand inequities in urban areas.

The relatively small sample size of our surveys in Hanoi and Dhaka have limited the inferences that can be made from our study. The samples will allow us to assess feasibility and appropriateness of the methods, but are not sufficient to derive estimates of depression or injuries.

Gaining agreement across government decision-makers on the content of the household survey may be challenging, particularly inclusion of mental ill-health and injuries. This will be mitigated through our long history of successful engagement with government together with our systematic review evidence of NCD prevalence. Changes in personnel in local government throughout the project could undermine attempts to work with staff to use and present data. The involvement of three cities in the study allows for some contingency if the potential risks do impact on data collection.

## Partnerships and collaboration

This is a collaborative project which includes research organisations from the UK (University of Leeds and University of Southampton), Nepal Health Research and Social Development Forum-international, Nepal; Centre for Injury Prevention and Research, Bangladesh and Advancement through Research and Knowledge Foundation, Bangladesh and Hanoi University of Public Health, Vietnam. Our approach to partnership is built on the principles of valuing expertise and differing contributions of our team members regardless of disciplinary or geographical location. For this reason, the lead researchers within each of the South Asian partners are named co-investigators on this proposal and will discuss and agree the strategic direction of this programme of work.

In Nepal, Bangladesh and Vietnam, our partners will work closely in an ongoing research-policy partnership[74 75] with the Bureaus of Statistics to understand existing survey methods and the potential for change. Representatives from these organisations will be invited to the final knowledge-exchange meeting to understand and discuss the novel methods proposed. Representatives of municipalities will also participate, particularly to inform discussions on data visualisation and utilisation. The team will also engage and share our methods and findings with multilateral organisations and academics seeking to address these issues globally. This close engagement with national and international decision-makers will maximise the impact of our study.

## Ethics and dissemination

Informed consent will be sought from all study participants before any data collection. While Global Positioning System data will be collected as part of the pilot of survey methods, this will be at cluster level rather than individual household level, so there is no risk of loss of confidentiality. Consent will be taken from all individuals appearing in any photograph from the photovoice exercise. Ethical approvals have already been obtained from the national ethical review bodies: Bangladesh Medical Research Council; Nepal Health Research Council; Vietnam Medical Research and Ethics Committee, and from the University of Leeds Medical Ethical Review Committee. The findings of this study will be disseminated using different approaches as appropriate to the target audiences of policy-makers nationally and internationally, survey organisations and academics. Dissemination methods will include conference presentations, national workshops, engagement in global meetings and events and publication of findings in open-access, peer-reviewed journals.

## Current status

At the time of submission (May 2018), the survey in Kathmandu has been completed and the surveys in Dhaka and Hanoi are underway. No analysis has begun. The participatory and qualitative methods are part way through.

### Author affiliations
[1]Nuffield Centre for International Health and Development, Leeds Institute of Health Sciences, University of Leeds, Leeds, UK
[2]Health Research and Social Development Forum—International, Kathmandu, Nepal
[3]Helen Keller International, Kathmandu, Nepal
[4]Centre for Injury Prevention and Research Bangladesh (CIPRB), Dhaka, Bangladesh
[5]Advancement through Research and Knowledge (ARK Foundation), Dhaka, Bangladesh
[6]Centre for Population Health Sciences, Hanoi University of Public Health (HUPH), Hanoi, Vietnam
[7]Flowminder Foundation, Stockholm, Sweden
[8]WorldPop, Department of Geography and Environment, University of Southampton, Southampton, UK
[9]Department of Social Statistics, University of Southampton, Southampton, UK
[10]School of Medicine, University of Notre Dame, Fremantle, Western Australia, Australia

**Acknowledgements** The authors would like to acknowledge the Medical Research Council (MRC) Global Challenges Research Fund (GCRF) for providing a grant to conduct this important research. The authors would also like to acknowledge the city and government authorities in Hanoi, Dhaka and Kathmandu who are supporting the study in their cities.

**Contributors** HE conceived the study, all authors (HE, TE, JNN, TM, JH, SM, RH, DW, HVM, SB, DT, HW) contributed to the development of study proposal, HE and ANP led the drafting of this manuscript with inputs from all coauthors (TE, TM, JNN, JH, DW, CC, CT, SB, RB, SK, RD, SG, SM, SM, JF, RH, TF, SN, HVM, DMD, BN, DT, HW), all authors read and approved the final version.

**Funding** This work is supported by Medical Research Council (MRC) Global Challenges Research Fund (GCRF) grant number MR/P024718/1.

**Competing interests** None declared.

**Patient consent** Not required.

**Ethics approval** School of Medicine Research Ethics Committee, University of Leeds; UK, Nepal Health Research Council, Nepal; Bangladesh Medical Research Council; Bangladesh; Ethical Review Board for Biomedical Research, Hanoi University of Public Health, Vietnam.

**Provenance and peer review** Not commissioned; externally peer reviewed.

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
