## [Reviewer comments · BMJ Open]

This paper was submitted to a another journal from BMJ but declined for publication following peer review. The authors addressed the reviewers' comments and submitted the revised paper to BMJ Open. The paper was subsequently accepted for publication at BMJ Open.

(This paper received three reviews from its previous journal but only two reviewers agreed to published their review.)

ARTICLE DETAILS

TITLE (PROVISIONAL)	Improving household surveys and use of data to address health inequities in three Asian cities: Protocol for the Surveys for Urban Equity (SUE) mixed methods and feasibility study
AUTHORS	Else, Helen; Poudel, Ak Narayan; Ensor, Tim; Mirzoev, Tolib; Newell, James Nicholas; Hicks, Joseph; Cartwright, Christopher; Wong, David; Tait, Caroline; Baral, Sushil; Bhattarai, Radheshyam; Khanal, Sudeepa; Dhungel, Rajeev; Gajurel, Subash; Manandhar, Shradha; Mashreky, Saidur; Ferdoush, J; Huque, Rumana; Ferdous, Tarana; Nasreen, Shammi; Van Minh, Hoang; Duc, Duong Minh; Ngoc, Bao; Thomson, Dana; Wallace, Hilary

VERSION 1 – REVIEW

REVIEWER	Richard Rothenberg, MD MPH Georgia State University, USA
REVIEW RETURNED	19-Jun-2018

GENERAL COMMENTS	Numerous collaborators present an overview of an ongoing project to provide better estimates of the size and characteristics of populations living in unofficial urban conurbations in LMICs. They provide considerable background and considered evaluation of these issues in a most informative and scholarly way. The text, however, is more of a discussion of methods than a protocol. The use of WorldPop is explained, and, in the case of Katmandu, they describe how they will use gridded population sampling, open-access maps, and will compare onse-stage to two-stage sampling. They use the future tense. At the end of the article, they inform us that the survey is Katmandu is completed and the other two are underway. Perhaps this is just an editing issue, but the message seems to be that here are our ideas (good ones, actually), and now we've done it, and the reader has only a general sense of what was actually done. The same applies to the questionnaire itself. They provide a nice precis of what it will include, but they obviously must have put together an actual survey instrument (the survey was completed) which they do not share with the reader. It may be that I have an incorrect sense of what a protocol is. My concept would be that a protocol is a detailed outline of the procedures that the researchers will use. If I'm wrong, then this document serves well to provide a general idea of what they wished to do, and my disappointment is that I would really like to know the
--

	nitty-gritty details. The fleshing out of these good ideas seems to be the real value. In another vein, the outline of what will be in the survey (actually, what is in the survey) includes considerable personal and community information, but might be served by having more of a focus on the structural and external determinants and their relationship to individuals. In particular, it would be of importance to know how certain types of social determinants affect individuals. We know, for example, that there are economic disparities within disadvantaged groups, but the specific ways in which subjects are exposed to those disparities—information you can only get from them—appears to be missing. The ability to connect a disparity to a person's experience provides an analytic tool that may be stronger than the usual indirect effects that we measure through mediation analysis. Finally, the group devotes little space to analysis (one brief paragraph). In particular, the sampling scheme that they actually ended up using does not have reference to the manner in which they derive variance estimates, usually a critical part of sampling design. A reference is made to the ICC, and of course they are comparing one and two stage clustering, but the specifics of how they will combine clusters—especially with some of the ad hoc adjustments that will be required—to create point and variance estimates is lacking. I get the distinct impression that any site to which their approach might be applied would have unique aspects, and it is likely that a single protocol would not fit all. They do not say this explicitly, but this may account for the absence of detail. Nor is this a criticism. In fact, the flexibility required for this kind of work is crucial.
--	--

REVIEWER	Patrick Harris University of Sydney, Australia
REVIEW RETURNED	29-Jun-2018

GENERAL COMMENTS	This is a well written protocol that will provide important new information for global health knowledge. I have made some minor suggestions in the attached, but no major changes are required. I am not a statistician however so this will require a further review. My only major concern is your slippage between health planning and urban planning - these are not the same things and you need to clarify what exactly is your focus and why. Also some more detail on the qualitative analysis would be helpful especially what themes you will be looking for and the oddly phrased Framework Approach given more detail. The reviewer also provided a marked copy with additional comments. Please contact the publisher for full details.
--

VERSION 1 – AUTHOR RESPONSE

Reviewer: 1

Reviewer Name: Richard Rothenberg, MD MPH Institution and Country: Georgia State University, USA Competing Interests: None declared

Numerous collaborators present an overview of an ongoing project to provide better estimates of the size and characteristics of populations living in unofficial urban conurbations in LMICs. They provide considerable background and considered evaluation of these issues in a most informative and scholarly way.

The text, however, is more of a discussion of methods than a protocol. The use of WorldPop is explained, and, in the case of Kathmandu, they describe how they will use gridded population sampling, open-access maps, and will compare one-stage to two-stage sampling. They use the future tense. At the end of the article, they inform us that the survey in Kathmandu is completed and the other two are underway. Perhaps this is just an editing issue, but the message seems to be that here are our ideas (good ones, actually), and now we've done it, and the reader has only a general sense of what was actually done. The same applies to the questionnaire itself. They provide a nice precis of what it will include, but they obviously must have put together an actual survey instrument (the survey was completed) which they do not share with the reader.

Response: Thank you for your feedback. As a protocol, our paper details what we planned to do. Our findings papers will detail what we have been able to do. In light of this, we would prefer to keep the future tense within this protocol paper. We note from other protocol papers published in BMJOpen that the use of the future tense is the norm. It has taken some time for the paper to reach this stage, and therefore we have now completed the first survey in Nepal and have reported this as recommended in the author guidelines. Please note, that we plan to publish the findings of the feasibility work and survey results for each city, these will explain exactly what was done in the study.

All the aspects of the survey questionnaire along with their sources are presented in table 2. We have included a link to our website where further details of all survey methods and the questionnaire are now available.

It may be that I have an incorrect sense of what a protocol is. My concept would be that a protocol is a detailed outline of the procedures that the researchers will use. If I'm wrong, then this document serves well to provide a general idea of what they wished to do, and my disappointment is that I would really like to know the nitty-gritty details. The fleshing out of these good ideas seems to be the real value.

Response: Thank you for your suggestion. We have included further information on the participant selection for the one and two-stage sampling procedures (p.6). Due to the space limitations of the article we have struggled to include all the details in the paper, but have included a link to the survey planning and interviewer manuals which provide detailed information on all the novel methods used within the surveys in the three cities (p.7) .

Once the research is complete, our findings papers from the study will describe the details of what was actually achieved and any challenges and improvements noted within the novel survey methods.

In another vein, the outline of what will be in the survey (actually, what is in the survey) includes considerable personal and community information, but might be served by having more of a focus on the structural and external determinants and their relationship to individuals. In particular, it would be of importance to know how certain types of social determinants affect individuals. We know, for example, that there are economic disparities within disadvantaged groups, but the specific ways in which subjects are exposed to those disparities—information you can only get from them—appears to be missing. The ability to connect a disparity to a person's experience provides an analytic tool that may be stronger than the usual indirect effects that we measure through mediation analysis.

Response: As mentioned by the reviewer, we have included personal and community information. We have included migration and social capital (section 2) in the questionnaire along with the personal

information in section 1 (household schedule). In the section 2, we have included indicators of social disparities, as suggested by the reviewer. Such as- migration status of the individual and households (Q201-210), staying in rent or own house (Q211), Secure from eviction (q212-214), future migration plan of the household (Q215), whether the household member is in a social group or not (Q217), trust in the community (Q218-219), provision of support if required (Q220), and question about lending and borrowing (Q221). As explained in the paper, one of our objectives is to compare different methods of measuring wealth and poverty and assess their appropriateness to the urban context. This will provide valuable information on the economic determinants of health. A further innovation that we have described in the paper is the development of a tool for measuring the extent to which an area can be classified as a 'slum'. This will allow assessment of the environmental determinants on health.

Finally, the group devotes little space to analysis (one brief paragraph). In particular, the sampling scheme that they actually ended up using does not have reference to the manner in which they derive variance estimates, usually a critical part of sampling design. A reference is made to the ICC, and of course they are comparing one and two stage clustering, but the specifics of how they will combine clusters—especially with some of the ad hoc adjustments that will be required—to create point and variance estimates is lacking.

Response: Further details on the analysis have been added on page 12, including expanding and hopefully clarifying how we will a) compare our survey methodologies (statistically primarily via ICCs), and b) analyse our survey data, including how we will account for the complex features of the surveys' designs, and more information on how we will (as in effect a sub-study) assess the validity of using the somatic questions to identify depression.

I get the distinct impression that any site to which their approach might be applied would have unique aspects, and it is likely that a single protocol would not fit all. They do not say this explicitly, but this may account for the absence of detail. Nor is this a criticism. In fact, the flexibility required for this kind of work is crucial.

Response: As this is a feasibility study of the novel methods, we will assess how acceptable and appropriate the novel methods are in different contexts. This will allow us to draw conclusions on their appropriateness for use in different settings. The finding may be that the novel survey methods, poverty measures, area slum assessment tool and injury and mental health questionnaires can be used with little adaptation in different urban contexts, but as this is a protocol paper, these questions cannot be answered in this paper. Our findings papers will share our analysis of all the approaches and measures and their applicability in different contexts.

Reviewer: 2

Reviewer Name: Patrick Harris

Institution and Country: University of Sydney, Australia Competing Interests: None declared

This is a well written protocol that will provide important new information for global health knowledge. I have made some minor suggestions in the attached, but no major changes are required. I am not a statistician however so this will require a further review. My only major concern is your slippage between health planning and urban planning - these are not the same things and you need to clarify what exactly is your focus and why. Also some more detail on the qualitative analysis would be helpful especially what themes you will be looking for and the oddly phrased Framework Approach given more detail.

Response: Thank you very much for your feedback. There are really helpful. We have now provided further information in analysis section on p.11. Regarding the health planning and urban planning, the words health and health-related have been added to planning, to differentiate from urban planning more broadly. It should be noted that in the three countries where the study takes place, urban health is the responsibility of the local government, not the Ministry of Health. So health planning is carried out by local government who are also responsible for urban planning. This has been clarified on p.10.

VERSION 2 – REVIEW

REVIEWER	Richard Rothenberg MD MPH FACP Georgia State University, USA
REVIEW RETURNED	14-Aug-2018
GENERAL COMMENTS	The reviewer completed the checklist but made no further comments.